# Securing Big Data Integrity for Industrial IoT in Smart Manufacturing Based on the Trusted Consortium Blockchain (TCB)

**Mazen Juma [1],\* , Fuad Alattar [2] and Basim Touqan [1]**

[1] Faculty of Engineering & IT, The British University in Dubai, Dubai P.O. Box 345015, United Arab Emirates; basem.tuqan@buid.ac.ae

[2] Siemens Industrial LLC, Dubai P.O. Box 33734, United Arab Emirates; fuad.alattar@siemens.com

\* Correspondence: mazen.juma@ieee.org

**Abstract:** The smart manufacturing ecosystem enhances the end-to-end efficiency of the mine-to-market lifecycle to create the value chain using the big data generated rapidly by edge computing devices, third-party technologies, and various stakeholders connected via the industrial Internet of things. In this context, smart manufacturing faces two serious challenges to its industrial IoT big data integrity: real-time transaction monitoring and peer validation due to the volume and velocity dimensions of big data in industrial IoT infrastructures. Modern blockchain technologies as an embedded layer substantially address these challenges to empower the capabilities of the IIoT layer to meet the integrity requirements of the big data layer. This paper presents the trusted consortium blockchain (TCB) framework to provide an optimal solution for big data integrity through a secure and verifiable hyperledger fabric modular (HFM). The TCB leverages trustworthiness in heterogeneous IIoT networks of governing end-point peers to achieve strong integrity for big data and support high transaction throughput and low latency of HFM contents. Our proposed framework drives the fault-tolerant properties and consensus protocols to monitor malicious activities of tunable peers if compromised and validates the signed evidence of big data recorded in real-time HFM operated over different smart manufacturing environments. Experimentally, the TCB has been evaluated and reached tradeoff results of throughput and latency better than the comparative consortium blockchain frameworks.

**Keywords:** IIoT; big data; blockchain; integrity; smart manufacturing; cybersecurity; Industry 4.0





## 1. Introduction

One of the fourth industrial revolution goals aims to shift manufacturing paradigms from automation to smartness due to the vast demand for quality, productivity, safety, efficiency, sustainability, and reliability in various industrial domains [1]. The deep-seated Industry 4.0 transformation is to integrate the industrial Internet of things capabilities into operations and production environments to foster interconnectivity, improve real-time monitoring, and empower control [2].

Since decades ago, automated manufacturing has evolved from monolithic proprietary systems to decentralized smart systems. It swiftly embraces the industrial Internet of things (IIoT) to collect big data at an ever-increasing rate from smart objects and monitored systems [3]. IIoT technologies provide sharper computing speed, advanced big data analytic capabilities, and cost-effective maintenance of proactively and remotely industrial infrastructures, leading to valuable business results [4].

In the industrial IoT era, big data integrity is the fundamental requirement of successful smart manufacturing that transformed from passively monitoring and controlling processes to improving overall operational effectiveness, acquiring big data in real-time, immediately accessing analysis outputs, and enabling on-the-spot actions anytime and anywhere [5].

Regularly, the big data of smart manufacturing is extracted, captured, and aggregated from different origins of smart industrial objects connected via IIoT networks at high speed with huge size to move through multiple manipulations, analysis, and interpretation processes stack along the integration cycle ended with greater outcomes [6].

These large-scale data are exchanged autonomously among machines, usually suffering from integrity issues when attacked or compromised during transmission from sources until they reach their destinations. Big data corruption leads to incorrect industrial controls and, therefore, improper engines of decision-making that cause a significant threat to the whole manufacturing value [7].

Industrial big data integrity is a serious concern for smart manufacturing. First, it is a substitute for industrial improvement and business profit and influences the national economy [8]. The latter is a research domain that implies more investment in digital technologies, modern concepts, and innovative methods to manage the v-dimensions constellation of the industrial IoT big data wisely and efficiently, practically volume and velocity.

The volume dimension, the foundation of big data, refers to the initial size and quantity of data collected. It is defined as massive data sets constantly being generated. In the industrial environment, it is not mainly created by human interactions but also by machines, networks, and smart objects, resulting in an enormous volume of data being analyzed [9].

The other dimension is the high-velocity data, such as IIoT sensor data streams, that have a high rate at which new data is generated and flows in from various sources for made available for analysis in real-time using data processing methods. With the vast and continuous growth, this real-time data is effectively managed to deliver more insight; thus, more value is created faster in smart manufacturing [10].

Nevertheless, the high architectural complexity of both dimensions presents a barrier for industrial environments to harness and analyze all the IIoT big data that is gathered. Handling the volume and velocity is challenging to address issues related to the potential of big data integrity [11].

Despite its multidimensional complexity, ensuring integrity automatically creates a clear path showing how big data has been used over time and its origins and becomes much easier and more reliable. Additionally, using suitable solutions, tracking down the single block of big data, and diverting significant industrial resources to sidestep all the disturbances to fulfill subject access requests without disrupting the big data pipeline [12]. At the same time, the (CIA) triad forms the foundation of data security and denotes the three primary pillars: confidentiality, integrity, and availability. Confidentiality indicates protecting against unauthorized disclosure of sensitive data. Integrity means ensuring that data cannot be tampered with or modified without authorization. Availability implies safeguarding authorized access to the data when needed [13].

As a second pillar of the above security triad, IIoT big data integrity interrelates with concepts of trustworthiness, consensual understanding, and appropriateness for use. It is determined by the originality of the data, the responsibility of the data source, and the standardization of the data usage [14].

The impacts of big data integrity improve the overall productivity of smart manufacturing through connected edge computing devices, industrial technologies, and diverse participants based on IIoT. However, there are challenges in ensuring the integrity of these data. One solution is to use blockchain technology to address these challenges and secure big data integrity [15].

In the course of this research, the trusted consortium blockchain (TCB) benefits from a secure and verifiable hyperledger fabric modular (HFM) to leverage trustworthiness in heterogeneous IIoT networks and guarantee the integrity of big data with high transaction throughput and low latency [16]. The proposed framework also has the ability to monitor and validate big data recorded in real-time, and experiments have shown that it performs better than other comparable blockchain frameworks.

## 2. Background

### 2.1. Smart Manufacturing Ecosystem

A smart manufacturing ecosystem has three overlapped perspectives: a smart factory, a digital thread, and a value chain. The smart factory perspective lines up with the purposes of the industrial internet of things, while the digital thread perspective is associated with the objectives of model-based manufacturing. Additionally, the value chain perspective supports the aims of the connected organization [17].

All these perspectives have substantial components that intersect in one focal part: operations management. Such a component orchestrates resource allocation, optimizes production processes alongside engineering specifications, and delivers a real-time flow of big data to improve supply chain functionality, as illustrated in Figure 1.

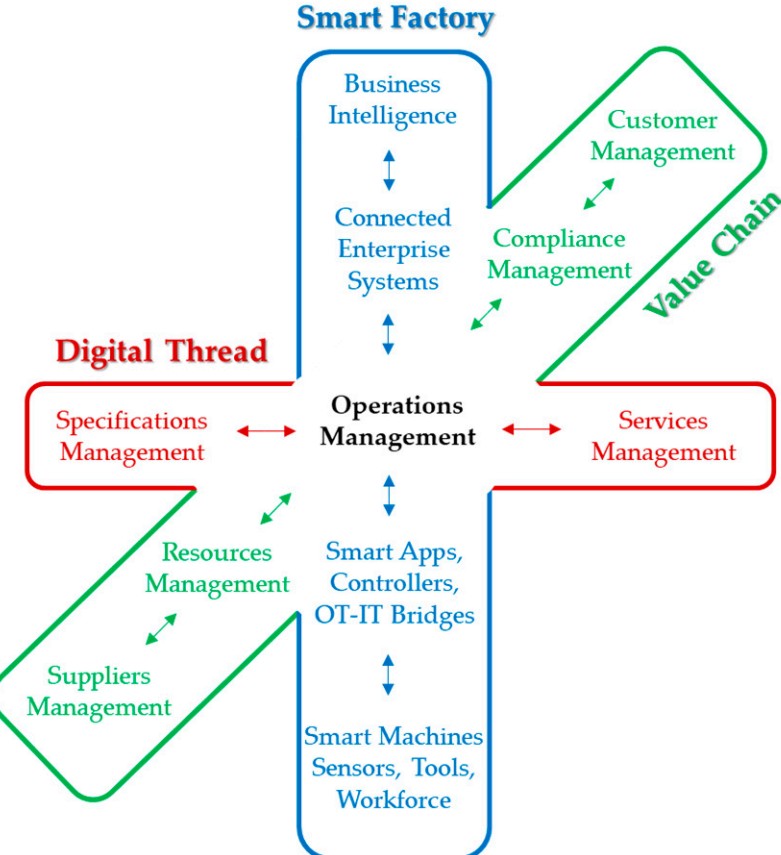

**Figure 1.** Smart manufacturing ecosystem.

At a glance, the components of the smart factory perspective include business intelligence to obtain cyclic updates of gathered big data for business goals and performance metrics. Connected enterprise systems to maintain real-time synchronization over big data interactions with used resources, the status of the production processes, and manufactured products [18].

The smart controllers and OT-IT bridge operations technology and information technology by exchanging big data immediately between systems and machines to help ease supervision, inspection, and maintenance duties. The final component presents the interactions of smart devices and the workforce via integrated systems and structured communications feeding simultaneous big data of production processing status [19].

In the second perspective, the digital thread components involve specifications and product service management. The specifications management fabricates product variations, process configurations, and engineering practices [20]. Meanwhile, product service management maintains the product lifecycle with big data gathered from process performance, adaptations, and substitution of products.

The third perspective is value chain management, which focuses on customers, compliance, resources, and suppliers. Customer management explores the needs and expectations of stakeholders regarding product requirements, orders in process, and change requests approval. The compliance management component provides institutional guidelines, administrates auditing, and monitors performance internally and externally [21].

Furthermore, resource management keeps the equipment and systems up and running based on vital capabilities and technical configurations. Last, supplier management identifies and establishes the supply chain to coordinate partners and sustain an acceptable level of quality [22].

### 2.2. Big Data Integrity

Big data integrity necessitates attaining the key principles of attributable, legible, contemporaneous, original, and accurate (ALCOA). Big data attributes reveal who and when it is observed and recorded, while legibility indicates an easy permanent understanding of the initial preserved big data [16].

The contemporary principle ensures that the big data is recorded as extracted and executed. The original form of the big data should also be sustained and accessible. Lastly, big data is considered accurate if it conforms to the protocols and is error-free, as depicted in Figure 2.

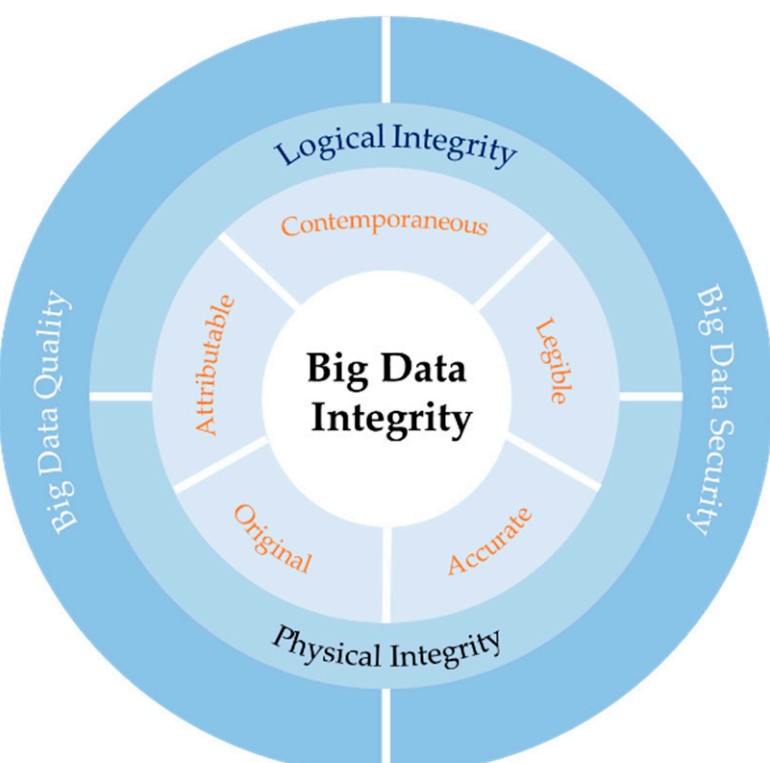

**Figure 2.** Big data integrity.

On the other hand, managing physical and logical types of big data integrity entails understanding the overall methods that enforce ALCOA principles in big data sets relationally and hierarchically. Hence, physical integrity keeps the stored and retrieved big data unchanged against several kinds of disasters, especially natural ones [23].

In contrast, logical integrity protects big data from being compromised by intentional or unintentional actions of humans. It is separated into four categories: entity integrity, referential integrity, domain integrity, and peer-defined integrity.

Entity integrity depends on creating unique values as the primary keys to identifying big data pieces to ensure they are not recorded more than once in various relational systems [18]. Next, referential integrity concerns the rules and constraints embedded into

structured big data to eliminate duplicate entries, occur proper changes, and guarantee that it is stored accurately and used uniformly.

As well, domain integrity is a set of acceptable measures and controls applied to each piece of big data to confirm the accuracy, limit the format, filter the type, and check the number of values allowed to be recorded in a particular domain [19].

At last, peer-defined integrity comprises the procedures and restrictions designed by the stakeholders to meet their precise prospects. However, these categories of logical integrity, in addition to physical integrity, must be considered and encompassed to safeguard big data integrity [24].

The outer shell in Figure 2 shows big data quality and security aspects, which play crucial roles incorporated with big data integrity to achieve optimum big data analytics and reach smart manufacturing goals. Big data security employs cohesive packages of systems, methods, and measures that protect big data from unauthorized access and prevent it from being corrupted over time.

In turn, big data quality moves further by encompassing an assortment of tasks and practices to govern the reliability, relevance, completeness, and maturity of collected, stored, and transferred big data [21].

### 2.3. IIoT Trust Styles

IIoT is the core pillar in Industry 4.0 that integrates industrial operations and information technologies to connect heterogeneouscyber–physical objects and communication protocols in the context of smart manufacturing. Thus, realizing, processing, and exchanging IIoT big data in real-time between machine-to-machine networks and business decisions saves capital expenditures and production expenses [25].

Although industrial data is critical to boosting smart manufacturing operations, three aspects influence the IIoT big data integrity, including the growing number of industrial things incessantly, storing and analyzing big data in a decentralization manner simultaneously, and end devices mobility with unstable interlinks in IIoT environments [23].

Based on that, trust among peers across IIoT systems is undoubtedly the most significant factor in securing big data integrity against complicated malicious activities depending on several indirect and non-measurable parameters.

However, IIoT trustworthiness is the combination of operational technology (OT) and information technology (IT). Additionally, it splits into five trust-based styles: behavior-based trust, computation-based trust, reputation-based trust, honesty-based trust, and accuracy-based trust, as shown in Figure 3.

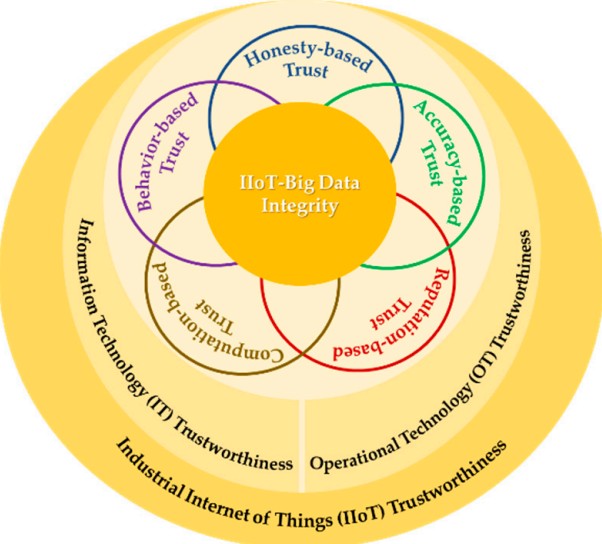

**Figure 3.** IIoT trust styles.

The first style of IIoT trust is based on the expected behavior of peers in the industrial network. This style considers the state of any peer has trustworthiness if it behaves as anticipated, even if its predictable behavior is not constant over time [24].

Secondly, computation-based trust is provided by technology from side-to-side computational techniques. It evaluates trustworthiness among IIoT devices within computing proximity and peer group interactions.

Thirdly, reputation-based trust is the accepted status of one peer on the basis of mutual IIoT big data from other peers in the entire network. This joint IIoT big data builds on a permutation of current observations and the historical statuses of the peers within a definite time and particular IIoT environment [25].

The fourth IIoT trust style is the honesty-based trust concluded evaluation of peer recommendations. Suppose industrial big data acknowledged from any peer matches expectations within a given temporal time or spatial situation; as a result, the recommender is considered an honest peer [26].

The last style is accuracy-based trust. This style designates IIoT big data integrity throughout industrial networks with limited resources. Peers could be accurate if their actual big data deployed at various locations of the heterogeneous IIoT environments lies within strict thresholds of trustworthiness [27].

*2.4. Consortium Blockchain Stack*

Blockchain is generally a decentralized peer-to-peer network that provides a shared ledger with continuous growth. The transactions of peers are chained jointly to create a block of records. Every peer in blockchain technology has a couple of public and private keys for contracting and confirming transactions [28].

The hash algorithm of Cryptographic SHA-512 is regularly applied to produce the hash values from the public key as the identity for each peer. The peers in the blockchain network run decentralized consensus algorithms to grant added transaction order validity over the blockchain ledger [29].

The first block of such a ledger, called the genesis, is developed by random transactions that are embedded and hard-coded into the blockchain system. Each block in the chain has two parts: upper and lower. The upper is the header, and the lower is the body.

The block header stores the cryptographic root hashes of the timestamp, the nonce, and the version to link the recent and prior blocks concurrently and build a secure chain of blocks. The block body comprises a list of executed transactions [30].

Typically, there are four distinct types of blockchain technologies. (1) Public blockchain is a permissionless process with no central authority. (2) Private blockchain is a permissioned process controlled by one authority. (3) Hybrid blockchain is a permissionless process controlled by one authority, blending critical elements of public and private blockchains. (4) Consortium blockchain is a semi-public and semi-private process controlled by a pre-selected group of equivalently permissioned peers [31].

The consortium blockchain eliminates the single-point vulnerability of the private blockchain depending on the decentralized network of multiple peers in trust of consensus for decision-making. Since carefully selected peers are permissible to authorize transactions, motivations are unnecessary for this kind of network, unlike in the public blockchain [32].

More precisely, the pre-selected group of peers in the consortium blockchain benefits from the public blockchain in facets of scalability and efficiency in parallel with permitting monitoring and central safeguarding, partially similar to the private blockchain [33].

Upon that, the hyper ledger fabric as a consortium blockchain application is designed to fit the requirements of collaborating peers exploiting the consortium blockchain to improve the smart manufacturing ecosystem. The consensus peers of the consortium group are identified and reputable; consequently, malicious peers will be prohibited from joining the consortium network easily [34].

As demonstrated in Figure 4, the consortium blockchain stack has four levels ascending from the bottom level of IIoT big data to the top level of application, passing by consensus

and contract levels, respectively. At the level of IIoT big data, the significant computations of the original big data-intensive and critical time are integrated with the physical edges of smart manufacturing processes. The captured big data from the smart controllers is considered the industrial big data source of the consortium blockchain [35].

**Figure 4.** Consortium blockchain stack.

The consortium blockchain stack continues to accomplish the consensus level by synchronizing the semi-public and semi-private consensus states for the group of peers, publishing industrial big data that demands to be scoped globally, and thus making local decisions for peer-to-peer networking to achieve the resiliency of the smart manufacturing environments [36].

At the contract level, the mapped consensus operations are implemented into the smart contracts that execute on a consortium blockchain to utilize the smart manufacturing services and resources. The decentralized methods dynamically organize smart contracts to settle smart manufacturing tasks.

Finally, the application level encapsulates the smart contracts to the programming manufacturing resources to assist the peers in decision-making regarding the programming manufacturing services [37].

## 3. The Research Gap

Industrial big data has great power to form significant value from smart manufacturing ecosystems in the present and future. The combination of the giant volume and high velocity of big data captured and extracted from overabundant origins with excessive, erroneous, and diverse levels of importance becomes more complicated along the essential analysis cycle, and it is inversely proportional to developing valuable insights into industrial big data [38].

Accordingly, the IIoT big data integrity challenges are associated with the volume and velocity dimensions of the big data domain in IIoT critical infrastructures. These challenges are divided into twofold: real-time transaction monitoring through data processing and peer validation at the destination of this process [39].

The challenge of real-time transaction monitoring for big data integrity stems from using the same infrastructure for big data analytics simultaneously with real-time transaction monitoring. However, the related security objects generate many false positives and alerts that require additional analysis since big data has been treated across several process phases [40].

The monitoring process continuously tracks and analyzes transactions in big data streams to secure their integrity. It adds automatically detects anomalies, suspicious patterns, and potential security risks. It also quickly identifies and responds to threats to protect against tampering, replication or transfer, or error checking [41].

Maintaining accuracy and consistency over a real-time monitoring lifecycle is crucial for preventing compromised big data from leading to profound losses. Securing big data integrity depends on protecting the industrial infrastructures and using comprehensive sight to improve the security of other systems. It is not sufficient to avert compromised big data at the originating process but conserved all over the entire process in place at high speeds [42].

Continuous transaction monitoring is a challenge that necessity be addressed. However, it causes several alerts and false positives that require further analysis. As the volume and velocity of big data increase, real-time monitoring problems also escalate because the processing sequence of big data is not followed or treatments are not properly or fully completed. It is hindered even when the process sequencing is respected but when malicious activities are involved in big data treatments [43]. In this manner, it is not adequate to set security controls to prevent corrupted big data from being transferred at the beginning stages of the data processing. The volume and velocity dimensions of big data crossing the protection rules and mechanisms of IIoT infrastructures need to leverage its integrity to be well-maintained and assured at every data process stage [41].

On the other hand, the challenge of the peer validation for big data integrity is not relevant to the large-scale data, but the loss of control and uncertain provenance throughout big data sources when the data creates. In this sense, the big data gathering phase is the initial input for data validation to ensure that the input source is not malicious to avoid fabricated or modified data from mingling with integrated ones [42].

Data validation is focused on both a state and a process, where data validation as a state means a big data set that is accurate. As a process, it mentions measures employed to confirm the accuracy of big data. It is essential for several reasons: recoverability, traceability, connectivity, stability, and reusability [42].

Still, it must go through various changes and processes to be useful for identifying relationships and making informed decisions. Effective enterprise security protocols include data integrity practices to ensure the validity and accuracy of data [43]. The input validation and filtering mechanism check the trustworthiness of big data origins and what the peer is responsible for. The overall score is computed and assigned to each big data provenance indicating the degree of big data trustworthiness. Based on the input score, the data is delivered to the next peer or rejected if it is insufficient [44].

Moreover, filtering malicious inputs is essential to prepare pure big data for the following data processing steps at each end-point across edge devices in the industrial internet of things to manage sound decisions related to the trusted peers where big data integrity is guaranteed [42].

The trusted peers validate all transactions; hence, peers are responsible for executing transactions by receiving requests and forwarding them to other peers for validation. A malicious peer executes the hijacked request against the consortium network, signs the response, and sends it back to the initial peer. The malicious responses are chronologically embedded into the transactions chain and channel-based big data block creation [45].

Each peer validates these transactions in the big data block and, holding a copy, if they are valid, adds the block to the local chain. The malicious peer has owned a copy of the big data block or hosted the entire local chain. Conversely, the peers' consensus among all big data blocks validates the transactions. Once validation is complete, all blocks receive an updated chain form [46].

## 4. Related Works

Many works studied more than a few aspects of the main technologies forming core layers of the blockchain-enabled IIoT big data in smart manufacturing. Nonetheless, limited

papers have been published dealing with issues securing big data integrity for industrial IoT using blockchain ledgers.

The next paragraphs briefly overview the previous research contributions presented on this paper topic as well as suggested approaches and technologies dealing with research gaps. The related works below have dissented into parts; the first part emphasizes the prior proposals to tackle the real-time monitoring challenge, and the second one states the exploration efforts to cope with the peer validation challenge [47].

For the first challenge, a number of the latest studies in [43–45] suggest well-organized designs of consortium blockchain based on modifications of Byzantine agreements [46,47] without offering big data integrity guarantees for the reason that they are designed for a closed network of trusted peers using a primary backup replication scheme to ensure fault tolerance. However, more or fewer proposals leveraged zero-knowledge protocols to withhold 'transactions' content [48], which involve expensive procedures for operating protocol and frequently trust an unattractive setup to bootstrap these protocols.

Moreover, the author of [49] revealed additional issues of big data integrity that were limited to specific technologies that did not have a holistic solution to improve efficiency and allow peers to share replicas with others in the network. As well, a selection of consortium blockchain projects in [50,51] does not address big data integrity, even though others [52,53] provide big data integrity with low performance comparatively. However, it tolerates decentralized control of a network of peers, where a separate security controls each peer. It uses a voting mechanism to reach a consensus.

Furthermore, consortium ledgers were proposed in the research performed by Refs. [54–56] depend on consensus protocols of Byzantine fault-tolerant for ordering the blockchain transactions. Hyperledgers in [57,58] uses practical Byzantine fault tolerance variants, which do not recall the algorithm of operating replications and, therefore, cannot appoint responsibility. This consensus mechanism combines small sets of delegates elected to validate transactions and reach a consensus.

The researchers [59] use the DiemBFT consensus protocol to secure the Diem blockchain established on HotStuff [60] and remove some attributes of big data integrity. In comparison, DiemBFT, based on the Tendermint consensus algorithm, is designed to be fast and more efficient. Alike, the Byzantine consensus in the works of [61,62] distributes trust among peers, while recent works on Byzantine fault-tolerant protocols focused on particular use cases to improve performance. One potential critique in [63,64] is that DiemBFT is a closed system, meaning that only authorized peers can validate transactions on the network, which could be less decentralized and more susceptible to censorship and manipulation than open networks. Additionally, it is still a new protocol and has not been tested wildly, so it is hard to handle large-scale transactions. Scalable Byzantine fault tolerance offers in [65] scale the peer replicas via a cryptography threshold to prevent big data manipulation during transmission via traditional voting, mechanisms to achieve consensus, and building big data blocks for various blockchains constructed on a decentralized consortium. Nevertheless, scaling to peer replicas typically deprives of increasing the blockchain scope, nor does it arbitrarily boost trustworthiness between heterogeneous peers [66].

Other papers have discovered misbehavior-based trust and its impact on IIoT big data integrity, such as the BFT2F protocol [67], which formalizes security and aliveness integrity after peers are compromised. However, it is insufficient because of susceptibility to dual spending cyberattacks.

Similarly, [68,69] issues an optional protocol after detecting accuracy-based trust, but it implements eventual big data integrity incompatible with consortium ledgers. It does not have the same level of security and big data integrity as a public ledger because the consensus mechanism used by consortium ledgers is typically centered on a smaller group of peers, which increases the consequence of a single point of failure or a malicious peer.

Correspondingly, the study in [70] prevents primary honesty-based trust from controlling the requests ordering of peers group. It did not address the multiple scenarios of honesty-based trust in one or two cases.

For the second challenge, the findings of [71] ensure that decentralized peers remain accountable for their transactions. It sustains high levels of overhead when utilized in a consortium ledger. It ensures the big data integrity of decentralized environments that all consortium peers agree on the state of this network, even in the presence of malicious or faulty peers. In contrast, [72] introduces procedures limited to Byzantine fault-tolerant for peer state replication and consortium ledger to improve the regular execution of transactions integrity. A consortium ledger is controlled by a group of pre-selected and trusted peers rather than being an open network. The combination of Byzantine fault tolerance and a consortium ledger improves the big data integrity of transactions so that all trusted peers reach a consensus within the consortium network.

Virtual peer accountability projected in [73] accomplishes integrity via checkpoints, although it has equal overhead performance, as [74] refers to the practice of holding oneself accountable to a peer group in goal setting. In SNP [75], a particular network implements peer accountability, recommending attribution of decisions for routing transactions. Such implementations progress the performance in specific areas merely were not precisely appropriate to consortium ledger.

Prosecutors [76] and BAR [77] incentivize peers to proceed with honesty-based trust by penalizing dishonest peers. The prosecutor model handles incentives to enhance performance, while BAR allows tolerance of three times faulty peers. Both mechanisms create a powerful deterrent for dishonesty. Peers in place are more likely to act honestly and transparently to avoid isolation. Conversely, peers did not improve the big data integrity when the incentives failed.

It enforces peers to proceed with honesty-based trust by penalizing dishonest peers. This approach creates a mechanism that encourages participants to be truthful and transparent in their actions. Additionally, it is placed on peers to complete their obligations honestly, and if any peer is found malicious, the peer is isolated [77].

Consortium accountability in [78,79] without behavior-based trust has been discussed before polygraph [80] and BFT protocol forensics [81] which proposition a consortium ledger for big data integrity mechanism but However, this assumption is based on the belief that those with a lower reputation in the trust system are not concerned about switching to a different peer group. These approaches propose various mechanisms for ensuring the integrity of big data in a consortium ledger, but they also have limitations. One limitation of polygraph testing is that it is inaccurate and produces false positive or negative results. Moreover, it is easily manipulated to rely solely on results in forensic investigations [82].

BFT protocol is limited to a high degree of trust among the network's peers, which is difficult to achieve in practice. Additionally, it is slow and inefficient, making it less suitable for high-performance consortium networks. Tendermint [83] is used in blockchain systems to ensure that all nodes in the network agree on the same data. However, it requires high trust among the validators' peers in the network, which is challenging, especially since it is not always very efficient in performance and is slow in certain conditions.

The other consensus algorithm is zero-lag BFT (ZLB) [84]. It combines the characteristics of BFT and a directed acyclic graph (DAG) to achieve faster and more efficient consensus. ZLB is restricted to relatively new and untested in large-scale industrial environments, making assessing its long-term reliability complex. ZLB requires many validators' peers to achieve its optimal performance. Tendermint and ZLB boost changes to the peer group; however, they act lesser than peers in the computation-based trust.

## 5. The Proposed Solution

In accordance with the given context above, the authors in this paper present a novel solution for bridging the research gap of IIoT big data integrity in smart manufacturing by coping with real-time transaction monitoring and peer validation challenges. The scope of the research contribution of this study is included developing, implementing, and evaluating the performance of the trusted consortium blockchain (TCB) framework.

As a proposed solution, the TCB is a decentralized trustworthiness framework that provides high-level security of industrial big data integrity for non-privileged peers and controls auditable computing over IIoT heterogeneous environments. It powers the hyperledger fabric modular (HFM) to achieve governance across a voting-based consensus and attain significant throughput and low latency performance.

*5.1. TCB Framework Design and Development*

The conceptual design of the TCB framework outlines three integrated layers consisting of the consortium blockchain layer wrapped with industrial IoT and big data layers to secure big data integrity. Each layer forked into seven integrated components. These components interact interoperability with a particular set of key functions to construct a trusted consortium blockchain framework.

As exemplified in Figure 5, the bottom layer of the TCB framework is the industrial IoT layer which involves core components: the IIoT equipment connector, IIoT device controller, IIoT unit communicator, IIoT edge transformer, IIoT big data accumulator, IIoT big data abstractor, and IIoT big data loader.

Alike, the middle layer of the proposed framework is the consortium blockchain layer which includes fundamental components: the CB store, CB provider, CB encoder, CB adaptor, CB controller, CB wrapper, and CB verifier. As well, the upper layer of the TCB framework is the big data layer which contains essential components: the BD access control enforcer, BD retriever, BD integrity detector, BD splitter, BD reconstructor, and BD integrity tracker. The following subsections show the extensive details of every component in the TCB framework layers.

5.1.1. Industrial IoT Layer

(1)  IIoT Equipment Connector: it establishes populated connections for industrial equipment, such as robots and remote actuators, with the required information. The associated facility sensors link targeted equipment to specific access points via field bus protocols. After granting access, industrial data collected from several pieces of equipment are metered and synchronized with pre-configured parameters.

(2)  IIoT Device Controller: it controls all IIoT big data produced by industrial equipment and manufacturing modulus and flows various devices, such as servo meters, embedded chips, PLC/PIDs, DCS, and CNC. Moreover, the control bus monitors the real-time data in unique source-based identification so that any part of industrial data is observed individually within a decentralized environment.

(3)  IIoT Unit Communicator: it provides an intermediary transmission to join physical controllers and computational edge transformers throughout 5G base stations and gateway nodes with high throughput. The RTUs bond between industrial data generated and broadcasted and handle communications using the M2M bus among decentralized devices over vast manufacturing areas to efficiently concentrate small levels of real-time data processing and transmit to central IIoT hubs.

(4)  IIoT Edge Transformer: it leverages the I/O senescing of smart edges attached to industrial equipment with rapid response time because of the low latency capturing and handling big data locally across IIoT hubs. The AMQP/MQTT servers acquired real-time data streams from various smart edges and standardized them to optimize the analysis of different security risks. Additionally, the OPC DA/UA servers are monitored the industrial data geographically for secure transmission depending on closing computing to the smart edges that produce the big data.

(5)  IIoT Big Data Accumulator: it delivers industrial data from storage nodes to the big data lake for initial pre-processing stages. The storage nodes module simultaneously supports multiple assemblies within the IIoT space, converts partially structured big data into fully structured ones through secure extraction techniques, and sends the processed big data into distinctive chunks. The big data lake achieves filtration, metadata reasoning, and fusing based on locale-time-sensitive.

(6)   IIoT Big Data Abstractor: it supports local big data aggregation acquired from heterogeneous manufactured sources and renders efficient real-time data with minimal delay by the substantial number of interoperability events within smart edges. It also manages quality levels of raw industrial data by trimming faulty, incomplete and duplicate big data to minimize the required resources and utilize the limited processing and transmitting capabilities.

(7)   IIoT Big Data Loader: it boosts the structured industrial data in historian repositories to improve rational big data computation and enhances decentralized loading capabilities. Afterward, the MES/MOM servers raise the readiness of abstracted IIoT big data by indexing, transferring, and storing the queries and responses among interconnected smart edges for loading directly to the consortium blockchain store in the middle layer of the TCB framework.

5.1.2. Consortium Blockchain Layer

(1)   CB Store: it stores designated big data fetched from historian repositories into decentralized storages and manages trusted access to them through the account authority rules. Moreover, it organizes the clean, complete, and error-free industrial data into small well-structured blocks with contextual metadata such as space, time, and location to detect big data integrity faults early.

(2)   CB Provider: it encapsulates designated big data into a hyperledger fabric modular (HFM) using testing and tran–chain interfaces. The testing interface receives main big data blocks from decentralized storage, constructs metadata mapping, and composers blocks structure in agreed formats. At the same time, the tran–chain interface detects and analyzes dual transactions to discover the malicious blocks. Both interfaces worked under the standardized policies of the contract governor to identify the correlation and control the chained big data blocks.

(3)   CB Encoder: it provides the fundamental requirements of formulating an encryption consensus in addition to customizing the standard contracts to run the consortium blockchain entry functions of given industrial data. Contract unifiers support these functions to normalize the peer-to-peer overlay networks. The contract testers also assess the encrypted blocks during peer consensus to identify errors and avoid vulnerabilities that lead to high exploits.

(4)   CB Adapter: it comprises diverse consortium blockchain interactions and builds cohesive capabilities for the chain–chain interoperability, including a registration chain, relay chain, and trans-gateway chain maintained by the trans-backbone chain. The standard API and tasks engine work together on the consortium blockchain to deliver essential adaptation to the cross-peer chain over the manufactured environments.

(5)   CB Controller: it employs identify manager to characterize the chains of industrial data blocks and discard the out-of-context ones. Likewise, the transactions manager ensures fast transmission via measuring time series and geolocations of peer chains. The data composers ordered assorted chains of big data blocks corresponding to the volume, speed, and period of chain creation to be ready for representation throughout big data interfaces.

(6)   CB Wrapper: it provides a participated multi-chain governor for registration, relay, and trans-gateway chains from the beginning of resources management to the end with permissions management and passing-by tasks management. These three critical mechanisms encompass concurrent focal points to administer the encryption peer consensus and big data integrity.

(7)   CB Verifier: it is a verification triad that jointly encompasses peers, credentials, and records verification. Verifying peers checks the structure of peering acting as a linking status. Thus, the verification of credentials confirms consortium consensus modeling. Additionally, verifying records proves the core consistency of in-line chains and off-chains of the big data blocks before enforcing them within access control in the upper layer of the TCB framework.

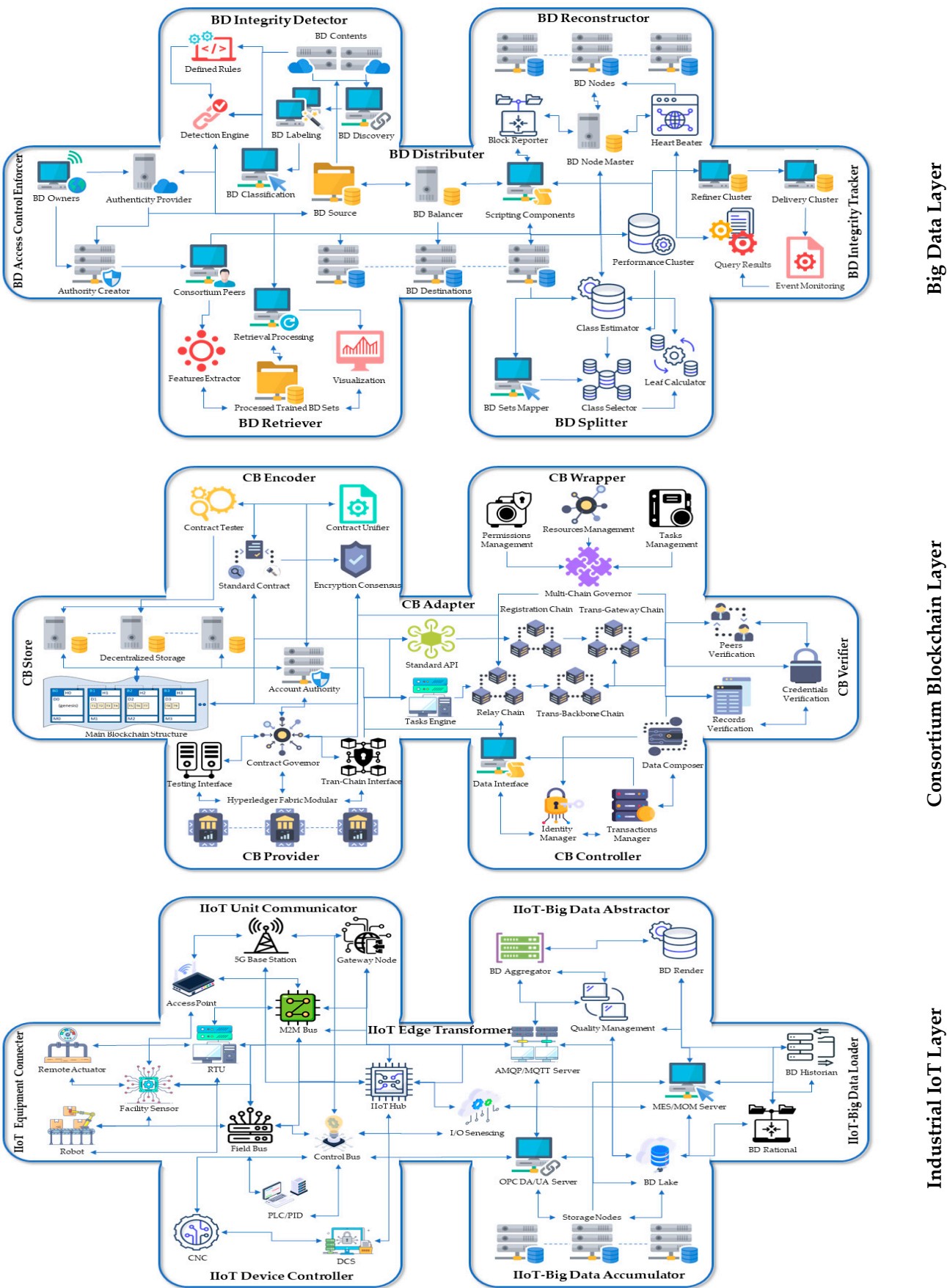

**Figure 5.** Architecture diagram of the trusted consortium blockchain (TCB) framework.

5.1.3. Big Data Layer

(1)  BD Access Control Enforcer: it is responsible for creating authenticity and authority between the big data owners and consortium peers. The authority creator enforces access control policies to all big data requests based on acceptable privileges granted to consortium peers. The big data owners seek and apply authorization rules afforded by the authenticity provider. Afterward, the big data integrity auditing logs are performed on the hyperledger fabric modular (HFM), and the big data blocks charge in the consortium blockchain.

(2)  BD Retriever: it retrieves the processed trained big data sets related to the requested big data blocks from the consortium blockchain using the retrieval processing. The features extractor merges and treats the integrity qualities of these blocks. Then, the retrieved blocks from big data contents are subject to a visualization course in order to prepare them for handling with the mechanisms of the big data integrity detector.

(3)  BD Integrity Detector: it analyzes the big data blocks to recognize the industrial data integrity aspects according to defined rules. Depending on the determined integrity level, big data blocks are discovered ahead of being labeled differently managing by the detection engine. Additionally, big data blocks are classified previous to placing into the big data source. The integrity metadata are generated during the detection analysis and held on the consortium blockchain to enable integrity capabilities.

(4)  BD Distributor: it assigns the big data destinations and maps them to the big data balancer. Formerly, it created scripting components aligned with the big data structure, hinging on the previously stated integrity preferences. Furthermore, these components send copies of the big data blocks from destinations to two distinctive tracks simultaneously. The first track is the big data sets mapper past the big data splitter, and the second is the block reporter passing through the big data reconstructor and then stored on the hyperledger fabric modular (HFM).

(5)  BD Splitter: it provides segmentation techniques for an additional coating of securing big data integrity. These techniques split big data blocks by class selector into integral and non-integral data sets based on specified integrity requirements. Next, the leaf calculator used checksum to ensure big data integrity by dint of SHA-512 encryption calculations for the original big data blocks. Then, the class estimator compared the hashing results to the initial encryption parameters after the performance clustering.

(6)  BD Reconstructor: it returns the big data blocks to their original forms using the integrity metadata saved in the block reporter over the hyperledger babric modular (HFM). The big data node master performs the segmentation and decryption to reconstruct the original blocks retrieved from big data nodes. The heart beater hardens the segmentation processing for low-integrity big data and decrypts the high-integrity portions of the big data blocks to avoid significant overhead measured by the performance cluster.

(7)  BD Integrity Tracker: it traces the streams of big data blocks delivered directly from the refiner cluster to the delivery cluster upon verified transaction queries of consortium peers or big data owners. Additionally, it leverages the event monitoring traceability on the basis of the termed thresholds and specific conditions provided by the hyperledger fabric modular (HFM). By doing this, the real-time execution of the active transaction queries shows continuous results during the industrial data integrity tracking process.

*5.2. TCB Framework Implementation and Deployment*

The prototype of the TCB framework is implemented based on two main algorithms that handle this research challenge of real-time transaction monitoring and peer validation, as mentioned in the following sections.

5.2.1. Real-Time Transaction Monitoring

Big data integrity is wrecked when the cycles of big data processing are not treated correctly. Furthermore, it is vulnerable even though the cycles are treated successfully, but malicious peers are aboard in these cycles. The suspicious behaviors of peers are not effortless to monitor, especially with big data's volume and velocity dimensions.

To cope with this challenge, examine every request that involves each big data transaction. Thus, assemble a compound metric of real-time transaction monitoring that yields two numerical values of the trustworthiness cycle and sequencing cycle.

The trustworthiness cycle monitors the peers who send and receive the transactions of the big data blocks at a certain time. Additionally, the sequencing cycle monitors the transactions of big data blocks that have been handled and accomplished within the respected orders before entering the analytical engine, as shown in Algorithm 1.

| **Algorithm 1.** Real-time Transaction Monitoring. |
|---|
| 01.      function RealtimeTransactionMonitoring(blockchain) |
| 02.          // Initialize a list to store suspicious transactions |
| 03.          suspiciousTransactions = [] |
| 04.          // Loop through all transactions in the blockchain |
| 05.          for each block in blockchain |
| 06.              for each transaction in block.transactions |
| 07.                  // Check if the transaction is suspicious |
| 08.                  if (isSuspiciousTransaction(transaction, blockchain)) |
| 09.                      // Add the transaction to the list of suspicious transactions |
| 10.                      suspiciousTransactions.append(transaction) |
| 11.                      // Notify the relevant authorities |
| 12.                      notifyAuthorities(transaction) |
| 13.                  end if |
| 14.              end for |
| 15.          end for |
| 16.          // Continuously monitor for new transactions |
| 17.          while (true) |
| 18.              newTransaction = getNewTransaction() |
| 19.              // Check if the new transaction is suspicious |
| 20.              if (isSuspiciousTransaction(newTransaction, blockchain)) |
| 21.                  // Add the transaction to the list of suspicious transactions |
| 22.                  suspiciousTransactions.append(newTransaction) |
| 23.                  // Notify the relevant authorities |
| 24.                  notifyAuthorities(newTransaction) |
| 25.              end if |
| 26.          end while |
| 27.      end function |

In this algorithm, the real-time transaction monitoring metric is calculated from when the transactions are sent to when they are treated based on the established steps. The transactions are leveraged throughout the trustworthiness and sequencing cycles in the correct order to manipulate the allowed or forbidden ones. Similarly, completing the authentication and authorization of the peers.

It allows transaction monitoring for malicious activity based on real-time analysis embedded architecture layers of the trusted consortium blockchain (TCB) framework to improve the accuracy of detecting suspicious and reduce false positives in industrial transactions.

Dissimilarity, traditional detection methods involve analyzing past transaction data to identify patterns or anomalies that indicate malicious activity. With real-time monitoring, potential malicious is detected and flagged as soon as the transaction is made, securing the integrity of big data and allowing for more immediate action.

In detail, big data owners communicate with the mutual authenticity provider and authority creator and rely upon TLS to establish secure channels via several interfaces.

These consortium 'peers' are verified against the hyperledger fabric modular (HFM) to certify and store the current state of the big data blockchain, whereas the security certificates are endorsed largely by the processed trained server.

The algorithm is statically programmed in C++ language and coded in Python scripts for easy reviewing and offers additional flexibility. It is stored and supported by the service API keys, passed command parameters, and defined rules.

The implementation uses positive payload as values and peer identifiers as keys. The HFM ledger authenticates the senders as registered peers implicitly. By rule, the authenticated peer identifiers are passed as a part of the first arguments in each command. The transfer commands take a receiver and a payload as added arguments.

Moreover, it checks that the sender has the correct checksum, then updates the states of both sender and receiver. The sequencing cycle confirms the uniqueness of these checks and updates, returning the existing states of sender and receiver peers.

The HFM ledger records a pair of updated key values, the first key checks that all transactions are valid and verify the correctness of signed transfers. The second one replays the peer authentication with supplementary signatures that passed as the final command argument.

To this end, the HFM ledger validates these signatures over other "transfers" arguments employing the sender has kept identifications before deploying the command. The received signatures are checked and overwritten their values by each transfer block. The updated and supported signatures are registered for the sender and receiver peers within the signed transactions concluded in the HFM ledger.

Signed transfers encompass further cryptographic handling for the transactions alongside the peers subject to their workloads and considerations. In such cases, the selected command codes are responsible for saving the signature values in the re-verified transfer context at adequate confidentiality. Eventually, the data block control returns the warning for every peer as the transfers executed have been compromised and start rolling back.

The warning explicitly contains the provisional serialization index of the compromised transfers in the HFM ledger. However, the HFM ledger runs parallel with the primary replication servers to broadcast, persevere, and immediately designate the rolled-back transactions. Later, peers query their 'transactions' selected state and use separate checksums.

Optionally, the HFM ledger correspondingly returns signed transfers for designated transactions, delivering self-governing evidence that the transfers executed at a serialization index in the HFM ledger created the returned transactions previously.

Likewise, when the HFM ledger fails to return signed transfers for several reasons, for example, big data node crashing, peer compromising, connection breaching, or time outing, the peer must still decide whether its transfers have been executed ahead of releasing new transactions. As a result, the HFM ledger records the definite sequence of big data blocks and unique peer identifiers in the transaction.

Furthermore, the peer tracks the up-to-date status of the signed transfers before submitting and receiving the existing position in the HFM ledger. Once the peer submits a new transaction, it is recorded after the previous position. The peer signature masks the transaction and its position in the HFM ledger, ensuring the restriction verifiability.

### 5.2.2. Peer Validation

The mechanism of peer validation fulfillments industrial data integrity by investigating the trustworthiness threshold of peers as big data sources and assigning a sub-score for each big data provenance that peers send or receive. After that, it computes and returns the total score of peer trustworthiness.

For that, implementing the peer validation algorithm checks the industrial data origination, defines who is responsible according to the acceptance or rejection of transactions passed to the other peers, and determines the big data integrity degree. These degree scores are recorded in the HFM ledger for additional integrity control.

The core originality of the peer validation algorithm is the dependency on the specific context process, in which a peer consortium is a source of industrial data; the peer behavior-based evaluation is vital to ensure the accuracy, reliability, and credibility of the big data sources.

Furthermore, this algorithm provides a more comprehensive evaluation of peer contributions to the consortium group and enhances the understanding of the overall behaviors implemented within the smart manufacturing environments.

The initial stages of the algorithm reduce big data volume by applying several filters belonging to trustful peers to avoid big data with low integrity degrees from transmitting into the next processing paces. Therefore, peer validation assured that the transactions based on the integrity degree scores were accurate and sound.

Furthermore, valid peers have access to the consortium blockchain, coded set in C++ and parameterized by python scripts stored in the HFM ledger. It comprises a range of functions such as deliver, request, vote, ack, and complete, as given in Algorithm 2.

These functions permit peers with acceptable trustful thresholds to review the HFM ledger, obtain valid peer identifiers, and verify their credentials. Additionally, it reinforces the restricted execution of 'peers' signed transfers on the HFM ledger for verifiability. Thus, the peers enable to reach consortium decisions, implement them transparently and update their primitives.

When one peer sends a transaction to a definite receiver, the transaction is restricted after running different checks on the basis of the selected updates on the HFM ledger. Their insertion index keys checks. The state of each check changes from alive to either reserved or passed. Apart from their states, checks are certainly not updated or ignored. After checking completely, the peers perform conditional voting on the verified transaction; each vote is updated as long as the transaction is still not recorded on the HFM ledger.

The acknowledgments of the consortium peers are witnessing the signed transfers running at the transaction index. These 'peers' signatures are stored in the HFM ledger, as the aliveness of the peer witnesses and their endorsements of comparatively topical states of the HFM ledger is useful feedback for auditing the checksum.

Upon that, the minimum number of required peer votes to pass the signed transactions is two-thirds of the total peer votes depending on its signed transfers. Peers call the HFM ledger to record their neoteric signed transfers. It checks at the outset that the new transaction is sent from an active peer with a well-formed scope.

Then, the HFM ledger adds a new big data block to the blockchain with an active peer identifier and valid credentials. It also updates present peer records by fixing their acceptance status as malicious. Hence, a single transaction needs multiple updates to ensure its one-off execution.

The HFM ledger records the fresh signed transfer for an active peer at the succeeding accessible index in the blockchain before returning it. Again, active peers call others to modify or record their final voting on the recently signed transfers. After reviews, the peer votes become non-adjustable and return true in approving the signed transfers.

The HFM ledger generally involves surplus checks on the status of the transactions. It enables the peer voters to check that the status of the active peer is set and meets the minimal requirements once the transactions are executed.

Likewise, when the peer voters are voting on adding a new active peer, their consortium decision is constituted based on accepted two third votes and passed before reaching a shared transaction index.

Any active peer calls request to compute the votes and decide if they be sufficient to pass transactions. It ensures that the transactions are still checked, tracks all their stored pre-status, and subsequently records on the HFM ledger to define the number of accepted votes required subject to the 'transactions' big data blocks.

---

**Algorithm 2.** Peer Validation.

---

```
01.     function PeerValidation(transaction, blockchain)
02.        isValid = True
03.        // Check if the transaction is already in the blockchain
04.        for each block in blockchain
05.            if (block.transaction == transaction)
06.                isValid = False
07.                break
08.        end for
09.        if (isValid)
10.            // Verify the transaction using digital signature
11.            if (verifyTransaction(transaction))
12.                // Check if the transaction is valid by comparing it to the current state of the
           network
13.                if (isValidTransaction(transaction, blockchain))
14.                    // Broadcast the transaction to the peer network
15.                    broadcast(transaction)
16.                    // Add the transaction to the local blockchain
17.                    addTransactionToBlockchain(transaction, blockchain)
18.                    // Notify peers of new transaction
19.                    notifyPeers(transaction)
20.                else
21.                    // Discard the transaction if it is invalid
23.                    discardTransaction(transaction)
24.                end if
25.            else
26.                // Discard the transaction if digital signature is invalid
27.                discardTransaction(transaction)
28.            end if
29.        else
30.            // Discard the transaction if it already exists in the blockchain
31.            discardTransaction(transaction)
32.        end if
33.     end function
```

---

Each transaction is executed entirely if its status succeeded in updating from check to pass; otherwise, the failed one updates from check to withdrawn. All aspects of the present and past transactions and their votes are stored in the HFM ledger to enable future auditing.

Regularly, consortium peers employ acknowledgments to confirm their contributions to the transactions and approve their states with verified signatures. It checks that these peers are not malicious, and the signed transfers properly update the index to the HFM ledger and do not overwrite signatures recorded previously.

Above all, accepted peers are a prerequisite to acknowledge their contributions to become active peers and, consequently, reward access to big data blocks stored in the verified transactions through the HFM ledger. Such acknowledgments require the presence of a supermajority of the acceptable peer voters.

On the other hand, deploying the TCB framework allocates the consortium peers to establish protected connection channels by exchanging the Diffie–Hellman keys. Additionally, it is pipelining the execution of transaction batches with sizes ranging between 800–1000 transactions for the IIoT LAN environment and 400–600 transactions for the IIoT WAN environment. The TCB uses the EverCrypt functions with SHA-512 encryption, verified Merkle trees, SECP-256k to secure signatures entirely, and the library of the MbedTLS for peering communications.

Furthermore, the checkpoints of big data integrity in the deployment area are founded in every 12,000 sequence number of industrial data blocks in the IIoT LAN networks and 6000 sequence numbers in the IIoT WAN networks, in addition to using the virtual smart

manufacturing benchmark with six singular peers who casually fulfill combinations of both types for transaction sources, such as orders and transaction actions as issues.

In addition, the benchmark entries size of the hyperledger fabric modular (HFM) varies between 200–350 bytes for each transaction entry, 250–300 bytes for every pre-prepare entry, 300–900 bytes for the single entry of prepare evidence, and 30–60 bytes for nonces entries depending on the transaction type.

*5.3. TCB Framework Evaluation Metrics and Testbeds*

At a glance, the performance evaluation of the TCB framework empirically provides a deep understanding of TCB capabilities to solve the real-time transaction monitoring and peer validation challenges. Upon that, this section presents a comprehensive demonstration of TCB evaluation metrics and testbed.

On the top, a set of evaluation metrics concludes that transaction throughput and latency measurements are used to assess TCB performance compared to existing consortium blockchain frameworks. The transaction throughput measurement over a dedicated cluster has quantified the units of transactions per second (tx/s) processed in a certain period and is measured by three metrics: average throughput, signed peer transfers, and ledger auditing.

In addition, the transaction latency measurement across the whole transactions in described experiments and calculating the time (ms) from transferring transaction to the consortium peer to receiving a consensus acknowledgment. Additionally, it is measured by three metrics involving average latency, 99th percentile latency, and IIoT network round trips.

The characteristic combination of the transaction throughput and transaction latency measurements with checkpointing for the experiments embraces three metrics: key value storing, functionality overhead, and checkpoint intervals. The results of these metrics for the proposed solution are compared against three popular solutions with open-source consortium ledgers as state-of-the-art baselines: IA-CCF [84], Pompe [82], and HotStuff [60].

In general, all conducted experiments were compute-bound, and collected results were around 10 runs with minimum errors. The TCB framework runtime testbed is configured and investigated suspiciously under diverse testing scenarios, with discrete cyberattacks fluctuating between compromises of big data integrity and inclusive testing environment crashes.

The experimental testbed setup is divided into three virtual environments based on SGX-enabled AWS VMs: dedicated cluster, LAN, and WAN networks, as illustrated in Figure 6. The dedicated virtual cluster consists of 4 trusted consortium blockchain servers, each one with an 8-Core 3.70 GHz CPU, Intel Xeon E-2288G, 32 GB RAM, and 60 Gbps NIC with full bi-section bandwidth; all of them run Ubuntu Linux 18.04.2 LTS.

Moreover, the local network in the AWS cloud includes four replica big data nodes with Fsv2-series VMs, 16-Core 3.40 GHz CPU, Intel Xeon E-8168, and 40 Gbps NIC links. On such durable configurations, the hyperledger fabric modular (HFM) with well-established BFT methods is expected to achieve the performance of a 10,223 (tx/s) as throughput, geth 8656 (tx/s), parity 64 (tx/s), and h-store 34,956 (tx/s), in addition to the external network with six consortium peers across 4 AWS VM pools from different locations.

The runtime testbed is ported to the LUA and EVM custom libraries of the mbedTLS with Everest in the virtual environments to terminate the internal TLS connections inside the VM clusters and deal with public-key certificates. For this reason, the HFM detects the TLS negotiations and the encrypted network traffic only. It uses suites of cryptographic algorithms for binary Merkle trees on the basis of SHA-256, ECDSA, ECDHE, and AES256-GCM.

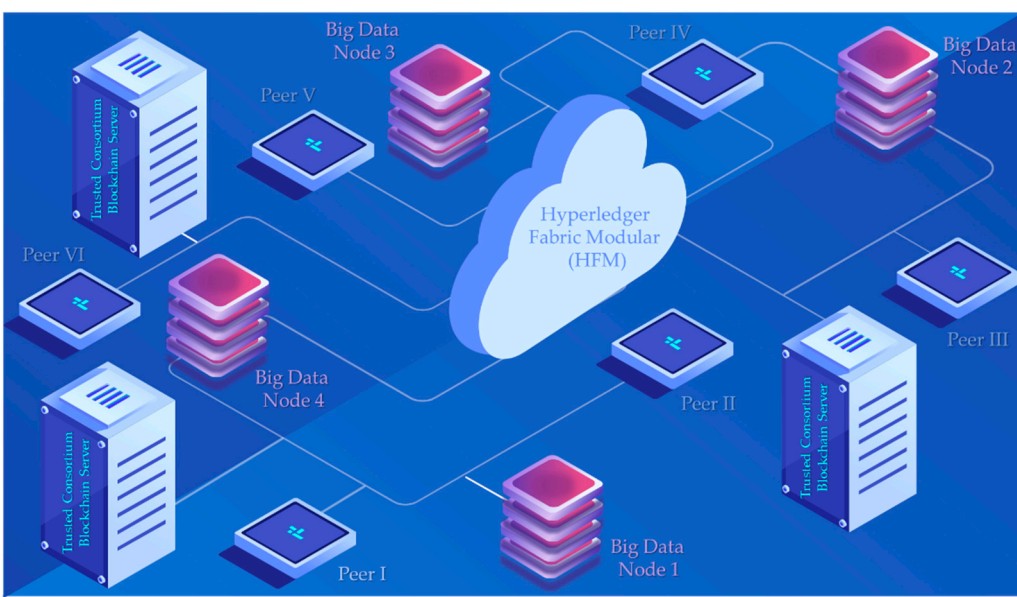

**Figure 6.** TCB runtime testbed.

Formally, the HFM observes the verified connections via JSON-RPC over the TLS module and forwards side-channel big data blocks and 'peers' identifiers using the TCB servers to handle the interfaces of big data node-to-node and peer-to-node communications with intermediate hashes caching and quick incremental up to 4M (tx/s).

## 6. Results and Discussion

The research results are presented in this section and described in detail through numerical tables and graphical charts. Additionally, the measured results are collected and analyzed precisely using testbed experiments.

The performance results of the proposed solution are evaluated, organized, and compared to the existing solutions over several testing scenarios with cyberattack events based on the selected metrics set of transaction throughput and latency measurements. Furthermore, the results are discussed, and the final research findings are interpreted in line with the research gap and related works.

In Table 1, the overall results show the transaction throughput measurement over the dedicated virtual cluster for the comparative solutions in this study. The TCB proposed solution has the best results in all evaluated metrics.

**Table 1.** Transaction throughput (tx/s) over dedicated virtual clusters.

| Comparative Solutions | Average Throughput | Signed Peer Transfers | Ledger Auditing |
|---|---|---|---|
| IA-CCF | 302,231 | 115,936 | 139,159 |
| Pompe | 487,644 | 204,775 | 330,459 |
| HotStuff | 313,697 | 128,897 | 162,711 |
| TCB | 508,444 | 217,661 | 367,119 |

It achieves 508,444 tx/s in average throughput metric higher than the Pompe solution, which reaches 487,644 tx/s, while the HotStuff and IA-CCF solutions have similar results, 313,697 tx/s and 302,231 tx/s, respectively, lower than the topmost ones.

The TCB and Pompe solutions also have higher throughput results in the the signed peer transfer metric of the HotStuff and IA-CCF solutions. The effectiveness of the HFM and HLF ledgers hired one-to-one in the TCB, and Pompe solutions played a crucial role in accomplishing 217,661 tx/s and 204,775 tx/s.

As illustrated in Figure 7, these throughput results are approximately double the throughput results of the HotStuff and IA-CCF in 128,897 tx/s and 115,936 tx/s. The influence on transaction throughput increases when the number of consortium peers in the AWS WAN environment spans multiple locations and decreases the correlated failures in IIoT networks.

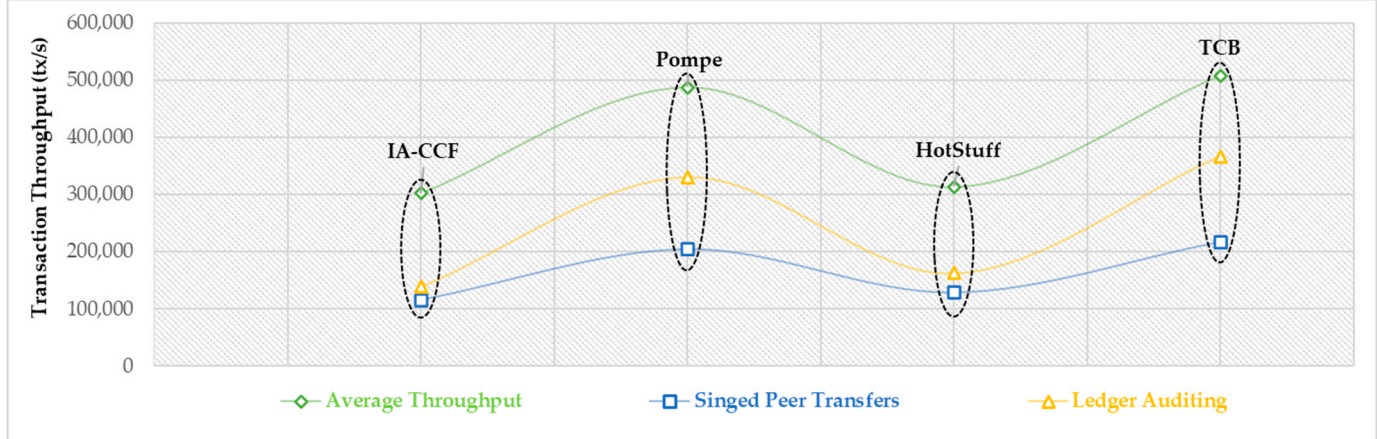

**Figure 7.** Transaction throughputs for comparative solutions.

In place of what was expected, the comparison against the TCB deployed in the AWS LAN environment reveals that the BFT consensus protocol of HotStuff is without key-value storing. The transaction throughput of the TCB grows with more peers because each peer confirms further signatures. Since each peer checks signed peer transfers in parallel, the transaction throughput rises whenever the peer number exceeds the number of big data nodes, which was four nodes in this runtime testbed experiments.

The above analysis also applies to the results of the ledger auditing metric, where the TCB achieves a throughput of 367,119 tx/s in the WAN environment, which is lower than its average LAN throughput. While the ledger auditing throughput of the Pompe reduces slightly to touch 330,459 tx/s with more peers, its throughput rests at 9% less than the TCB.

The throughput results of the ledger auditing for HotStuff and IA-CCF are even lower than TCB and Pompe since they execute cryptographic operations. When comparing auditing time to execution time, the auditing performance measures throughput at 162,711 tx/s for HotStuff and 139,159 for IA-CCF.

The ledger auditing of HotStuff is about 15% faster than IA-CCF because of the low overhead for ledger auditing writes and transaction signing. In every execution, HotStuff verifies one-sixth rather than up to one-third of new signatures. For IA-CCF, the performance gap of ledger auditing enlarges to 62% as more peers make connections and cryptographic workloads during the execution. The bottleneck of ledger auditing is the proof of peer transaction signatures that are parallelized insignificantly.

In Table 2, the conclusive results represent the transaction latency measurement upon the dedicated virtual cluster that contains several comparable metrics of average and percentile latencies and IIoT network round trips. The functionality of these metrics considers creating transaction workloads, the entry sizes of the ledgers fitting in the cache, and the consensus costs among peers with verifying their signatures.

**Table 2.** Transaction latency (ms) over a dedicated virtual cluster.

| Comparative Solutions | Average Latency | 99th Percentile Latency | Round-Trip Latency |
|---|---|---|---|
| IA-CCF | 188 | 198 | 215 |
| Pompe | 391 | 437 | 509 |
| HotStuff | 346 | 398 | 445 |
| TCB | 290 | 254 | 326 |

In the measurement of the average latency metric for comparative solutions under high loads, IA-CCF's average latency was 188 ms, which is nearly less than twice that of Pompe with 391 ms and HotStuff with 346 ms. For both solutions, the average latency is affected by the number of IIoT network round trips, whereas the peers receive transactions with their acknowledgments in four and five round trips for HotStuff and Pompe, respectively.

The TCB has second place in average latency next to IA-CCF with 290 ms due to being marginally impacted by the higher WAN pipelining latency compared to the LAN deployment. Consequently, the 99th percentile latency metric focuses on the maximum latency for the fastest 99% of transactions processed by the peer in a certain period.

As exemplified in Figure 8, the 99th percentile latency of the IA-CCF is the lowest one measured up to other results of all comparative solutions, including the proposed solution in this study; hence, the peers processed 99% of transactions in less than 198 ms. These results are consistent with the above results of the average latency metric for the comparative solutions.

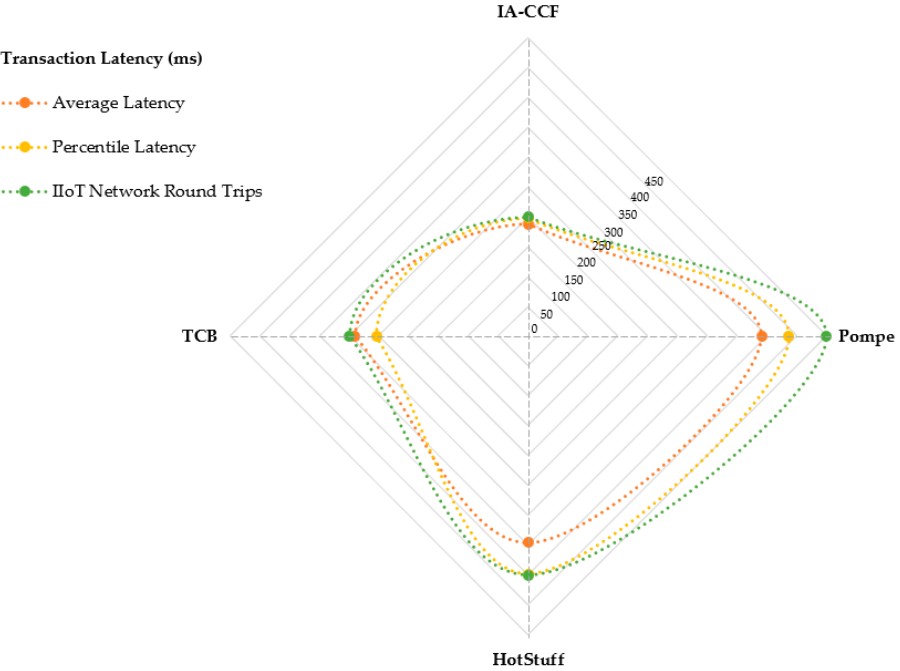

**Figure 8.** Transaction latencies for comparative solutions.

Again, the TCB attained the second rank in the results of the overall percentile latency following the IA-CCF solution with 294 ms. The TCB requires more time to verify transactions depending on the path length of the Merkle tree and the number of peer signatures needed to be checked at once.

Far away from the results of the first two solutions, the remaining ones have worse results in the percentile latency, 398 ms for HotStuff, and 437 ms for Pompe, since these solutions handled the number of transactions with bounded entry sizes available in the big data nodes due to the given path length of the IIoT network which enables the verification of entries between 250 and 900 bytes only.

Consequently, the key performance metric of the round-trip time has measured the duration from when a peer sends a transaction to when it receives acknowledgments from consortium peers in the IIoT network. It is a focal factor, and there are other metrics for measuring industrial network latency.

The whole cost of round-trip latency in the runtime testbed is overlooked by the peer signature verification, which usually takes 16 ms and 22 ms for each peer. The total results of the round-trip latency for the IA-CCF and TCB solutions are lower than the ones of the HotStuff and Pompe solutions. Thus, IA-CCF reaches 215 ms, and the TCB completes 326 ms in round-trip latency.

For context, the operating protocols of Byzantine consensus have similar functionality in HotStuff and Pompe frameworks; therefore, HotStuff's round-trip latency is 445 ms, and Pompe's is 509 ms. The TCB utilizes techniques for decreasing round-trip latency incomparable with HotStuff and Pompe solutions. All discussed results in altered fault tolerance to guarantee big data integrity, and demonstrate the efficiency of the TCB.

In Table 3, the transaction throughput/latency measurement evaluates the performance correlation between the throughput and latency measurement via three composite metrics, including key value storing, functionality overhead, and checkpoint intervals.

**Table 3.** Transaction throughput/latency (tx/s) with checkpointing.

| Comparative Solutions | Key Value Storing | Functionality Overhead | Checkpoint Intervals |
|---|---|---|---|
| IA-CCF | 57,579 | 11,118 | 53,209 |
| Pompe | 46,845 | 10,763 | 41,018 |
| HotStuff | 60,986 | 12,799 | 55,415 |
| TCB | 61,100 | 13,102 | 56,219 |

The TCB solution triumphs in the key value storing metric by 61,100 tx/s and maintains overhead functionality metric above 13,102 tx/s in parallel with touches 56,219 tx/s for checkpoint interval metrics. Therefore, the results of the proposed solution significantly excelled in the whole results for comparative solutions of all these metrics.

The performance of the TCB is impacted by changing the number of key entries processed and stored by the consortium peers. As expected, TCB throughput upturns when the number of key entries in the big data nodes buildups because the mapping implementation of the access time demonstrates the low cost of stored values and expands logarithmically with the number of key entries.

The throughput/latency results of the comparative solutions' key value storing metric vary upon the entries queueing delays. So, HotStuff's throughput/latency is 60,986 tx/s, which is lower than the TCB by about 1%. Additionally, IA-CCF has the third place in key-value storing with 57,579 tx/s turndowns, 5.7% of TCB performance.

Lastly, the Pompe solution exhibits an order of entry size dropper throughput/latency result of 23.3% to be 46,845 tx/s matched to the TCB results because all entries must be stored after signing, and the peer signs and stores the values for each entry in every big data node separately.

As illuminated in Figure 9, the overhead functionality results provide a clear perception of the 'ledgers' capability employed by comparative solutions to drive functionality within consensus protocols with peer accountability for generating transactions.

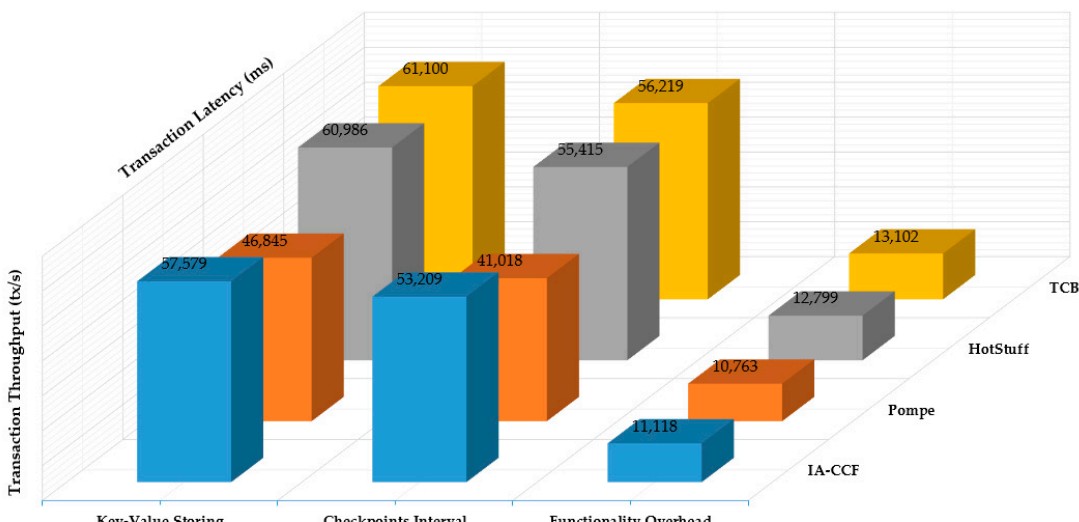

**Figure 9.** Transaction throughputs/latencies for comparative solutions.

The TCB achieved 13,102 tx/s, which explores the impact of fulfilling functionality overhead on its throughput/latency over the dedicated virtual cluster. By straightening out the transaction ordering and peer consensus, IA-CCF accomplishes a functionality overhead of 11,118 tx/s.

In contrast, the HotStuff solution uses signatures substantially to remove the functionality overhead of performing transactions against the transactional key-value storing rather than specifying consensus protocol in an attempt to double throughput with 12,799 tx/s. However, the results of Pompe indicate that its functionality overhead regularly comes from the cryptographic procedures entailed for validating peer transactions; as a result, it ends at 10,763 tx/s as the lowermost performance.

Finally, the results of the checkpoint intervals reveal their effect on the performance of the comparative solutions where the size of the key value storing for the checkpointing workloads and frequencies varies. Thus, the TCB solution has the best results rather toother solutions, with 56,219 tx/s.

As predictable, the results of the HotStuff and IA-CCF were convergent with 55,415 tx/s and 53,209 tx/s, respectively, for checkpoint intervals between 60 and 85K every 500 milliseconds and replay the transactions at the same time into proper practice.

The checkpointing overheads enlarge with the key values' sizes; hence, the checkpoint interval results of both solutions were close to the TCB. On the contrary, the result of the Pompe solution was noticeably the lowest one at 41,018 tx/s due to an asymmetric cryptographic procedure with two orders of magnitudes more than the TCB.

## 7. Conclusions

The fourth industrial revolution aims to shift manufacturing from automation to smartness to improve quality, productivity, efficiency, and sustainability. It has a deep-seated transformation involving integrating industrial IoT capabilities into operations and production environments to foster interconnectivity and improve real-time monitoring and control.

Smart manufacturing has evolved from monolithic proprietary systems to decentralized smart systems, which are now embracing IIoT technologies to collect big data at an ever-increasing rate. These technologies provide faster computing, advanced data analytics, and cost-effective maintenance of industrial infrastructures, leading to valuable business results.

Big data integrity is crucial for successful smart manufacturing. It implicates moving from passive monitoring and control to improving overall operational effectiveness, acquir-

ing big data in real-time, immediately accessing analysis outputs, and enabling on-the-spot actions anytime and anywhere.

This paper focused on two challenges of big data integrity in the industrial internet of things stemming from big data v-dimension complexity. It proposed an optimal solution to cope successfully with these challenges for securing big data integrity in smart manufacturing environments.

Furthermore, such research is developing, implementing, and evaluating the performance of the trusted consortium blockchain (TCB) framework to leverage the trustworthiness levels of the big data cycle through real-time transaction monitoring and govern peer validation.

The three layers of the TCB framework are integrated and built on top of the hyperledger fabric modular (HFM), which enhances high transaction throughput and low latency of the consortium peers' transactions in heterogeneous IIoT networks to manage, preserve, and accomplish ALOCA principles of big data integrity.

By experimentation, the TCB has been running on the configured testbed and examining carefully under multiple testing scenarios with different cyberattack mockups ranging from compromises to crashes. Additionally, the performance of the new framework is evaluated and visualized based on several metrics according to transaction throughput and latency measurements.

The TCB and Pompe solutions have an average throughput higher than the HotStuff and IA-CCF solutions. The TCB and Pompe solutions have higher signed peer transfers than the HotStuff and IA-CCF solutions. At the same time, HotStuff and IA-CCF are lower than the TCB and Pompe for ledger auditing results.

The average latency of IA-CCF is significantly lower than that of Pompe and HotStuff, although the 99th percentile latency of IA-CCF is the lowest among all comparative solutions. The round-trip latency for the IA-CCF and TCB solutions is lower than that of HotStuff and Pompe.

The TCB's performance is affected by the key value storing processed and stored by consortium peers and increases as the number of key entries in big data nodes increases. Other solutions, such as HotStuff and IA-CCF, have lower key value storing rates, while the Pompe solution has the lowest rate. The TCB performs the best in terms of functionality overhead, followed by IA-CCF, HotStuff, and Pompe. Lastly, the TCB has the highest rate in checkpoint intervals, while HotStuff and IA-CCF are close behind, and Pompe has the lowest rate.

The overall empirical results of the proposed solution have been discussed and interpreted compared to the results of the existing consortium blockchain frameworks. Significantly, the TCB solution achieves high throughput and latency results better than the comparative other diverse solutions with guarantees of industrial data integrity.

**Author Contributions:** Conceptualization, M.J. and F.A.; methodology, M.J. and B.T.; software, F.A.; validation, M.J. and B.T.; formal analysis, B.T.; investigation, B.T.; resources, F.A.; data curation, F.A.; writing—original draft preparation, M.J.; writing—review and editing, F.A. and B.T.; visualization, F.A.; supervision, M.J.; project administration, M.J. and B.T. All authors have read and agreed to the published version of the manuscript.

**Funding:** This research received no external funding.

**Institutional Review Board Statement:** Not applicable.

**Informed Consent Statement:** Not applicable.

**Data Availability Statement:** Not applicable.

**Conflicts of Interest:** The authors declare no conflict of interest.

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
