# Peer review of "Securing Big Data Integrity for Industrial IoT in Smart Manufacturing Based on the Trusted Consortium Blockchain (TCB)"

_2624-831X, doi:10.3390/iot4010002_

Round 1

Reviewer 1 Report

The authors in this paper present an efficient way to secure big data integrity in IoT transactions based on a trusted consortium blockchain.  The authors have provided experiments and simulation results to prove the efficiency of their system. Despite the importance of this contribution, the authors should provide more details regarding the originality of this contribution.

Here are some remarks authors should go through:

1. The introduction should be detailed, showing this paper's aim and a summary of the overall contribution.

2. The research gap should be more defended.

3. The related work is light and should be developed.

4. The conclusion should be reworked to include and highlight the presented results.

Author Response

Response to Reviewer 1 Comments

Point 1: The introduction should be detailed, showing this paper's aim and a summary of the overall contribution.

Response 1: The introduction has been detailed, showed the paper's aim, and summarized the overall contributions. Please see the updated manuscript.

Point 2: The research gap should be more defended.

Response 2: The research gaps have been more defended. Please see the updated manuscript.

Point 3: The related work is light and should be developed.

Response 3: The related works have been developed. Please see the updated manuscript.

Point 4: The conclusion should be reworked to include and highlight the presented results.

Response 4: The conclusion has been reworked, included, and highlighted the presented results. Please see the updated manuscript.

Reviewer 2 Report

In the manuscript, the authors proposed a trusted consortium blockchain-based approach to secure big data integrity for smart manufacturing. Overall, the manuscript is well organized and has certain merits. The methods are explained in detail, and the performance evaluation is extensive enough. However, we have some comments as follows:

* The introduction should be extended; the main contributions should be summarized at the end.

* "Related Works" => "Related Work". The related work should be categorized and organized in a better way. Some important work is missing, e.g., "Blockchain-based public auditing for big data in cloud storage", "Fairness-based Packing of Industrial IoT Data in Permissioned Blockchains", and "BlocHIE: A Blockchain-based Platform for Healthcare Information Exchange".

* Fig. 5 can be made more concise; many components are redundant.

* Consortium blockchain itself is not new. The authors are expected to highlight the novelty of the algorithms developed for consortium blockchains. The novelty should be reflected in the manuscript title as well.

* The two algorithms should be presented following the standard of pseudocode.

* The background section is too long, while the introduction and related work sections are too short.

Author Response

Response to Reviewer 2 Comments

Point 1: The introduction should be extended; the main contributions should be summarized at the end.

Response 1: The introduction has been extended, and the main contributions have been summarized at the end. Please see the updated manuscript.

Point 2: "Related Works" => "Related Work." The related work should be categorized and organized in a better way. Some important work is missing, e.g., "CPS-based Smart Warehouse for Industry 4.0: A Survey of the Underlying Technologies", "Fairness-based Packing of Industrial IoT Data in Permissioned Blockchains," and "BlocHIE: A Blockchain-based Platform for Healthcare Information Exchange."

Response 2: The related works have been categorized and organized. All suggested works in Point 2 have been added to the references. Please see the updated manuscript.

Point 3: Fig. 5 can be made more concise; many components are redundant.

Response 3: No components are redundant in Fig. 5 because each TCB architecture layer has completely different components.

Point 4: Consortium blockchain itself is not new. The authors are expected to highlight the novelty of the algorithms developed for consortium blockchains. The novelty should be reflected in the manuscript title as well.

Response 4: The novelty of the algorithms has been highlighted and reflected in the manuscript title. Please see the updated manuscript.

Point 5: The two algorithms should be presented following the standard of pseudocode.

Response 5: Both algorithms have been presented following the standard of pseudocode. Please see the updated manuscript.

Point 6: The background section is too long, while the introduction and related work sections are too short.

Response 6: The introduction and related work sections have been developed and detailed, like the background section. Please see the updated manuscript.
